# A Control Method Based on a Simple Dynamic Optimizer: An Application to Micromachines with Friction

**DOI:** 10.3390/mi14020387

**Published:** 2023-02-04

**Authors:** Leonardo Acho

**Affiliations:** Department of Mathematics, ESEIAAT-Universitat Politècnica de Catalunya, 08222 Terrassa, Spain; leonardo.acho@upc.edu

**Keywords:** regulation control design, dynamic optimizer, micromachines, LuGre friction model, finite-time stability

## Abstract

In Micromachines, like any mechanical system, friction compensation is an important topic for control design application. In real applications, a nonlinear control scheme has proven to be an efficient method to mitigate the effects of friction. Therefore, a new regulation control method based on a simple dynamic optimizer is proposed. The used optimizer has a finite-time convergence to the optimal value of a given performance index. This dynamic process is then modified to produce a new control scheme to resolve the regulation control statement. A stability test is also provided along with numerical simulations to support our approach. We used the Lyapunov theory to confirm the stability, in finite-time, of the obtained closed-loop system. Furthermore, we tested this controller in a scenario where the reference signal was a time-varying function applied to a micromachine with friction. Numerical experiments showed acceptable performance in mitigating the effects of friction in the mechanism. In the simulations, the well-known LuGre friction model was invoked.

## 1. Introduction

Micromachines for micro-position control using piezoelectric actuators have been an important technology in some precision engineering applications [1,2,3]. Piezoelectric actuators are commonly used in micromachine control due to their infinite displacement resolution and low heat generation, among other factors [1,3]. However, these actuators exhibit hysteretic friction, which plays an important role in control design [1,3,4]. Regarding this point, many control strategies have been proposed. Among these, we have those based on PID schemes [3], modern control theory [2], robust analysis [5,6], neuronal theory together with terminal sliding mode control [7], and so on. Here, we propose a new control scheme based on a kind of dynamic optimizer [8].

For several years, control design using optimal mathematical tools has been a well-known control theory approach [9,10,11]. Even so, among the different methods, the extremum seeking technique remains a current control methodology in some engineering applications [12,13,14]. This control approach has gained considerable popularity after the publication cited in [15]. Essentially, the extremum seeking control scheme was proposed as a tool for model-less stable dynamical systems [16]. It is well known that real-time optimization integrated into a control scheme can improve closed-loop system performance. Furthermore, it can also help stabilize an unstable system [17]. A disadvantage of this technique is that it requires a disturbance signal [13]. This limits the convergence of the system and complicates the adjustment of the control system parameters [13]. On the other hand, a dynamic optimizer with convergence in finite-time was given in [8]. This optimizer did not require any disturbance signal. Therefore, the main objective of this article was to adapt this scheme to a dynamic system and satisfy the regulation control problem statement. We used Lyapunov theory for the finite-time stability test. In addition, numerical experiments were carried out to support our main contribution. For the numerical experimentation we used the Xcos application of the Scilab software. Both regulation and tracking control issues were numerically analyzed. This includes the application of a micromachine with the phenomenon of hysteretical friction captured by the LuGre micromachine friction model, reported in [1]. According to numerical experimentation, our control approach presented an acceptable performance. In summary, the control design challenges are shown in Table 1.

Even though the technology is rapidly increasing in micronized devices, the phenomenon of friction is still present [18]. This happens because friction is highly dependent on the operating conditions of the system under control [18,19]. Furthermore, mitigating the effects of friction in microelectromechanical systems (MEMS) is still a great challenge [19]. On a small scale, friction is far from being represented mathematically. This is due, for example, to the fact that the normal force represented as a constant proportional to the ground reaction force is no longer fully fulfilled for micro devices [19,20]. In fact, at low microscales, friction is represented mathematically using an empirical law and also including an aging algorithm [21]. Furthermore, in some micromachines, the friction phenomenon exhibits hysteresis in a rate-dependent nonlinear behavior [22]. Therefore, a friction compensator that does not fully depend on friction modeling is a great challenge for micromachine control.

We report on contributions in regard to friction compensation for micromachines. Among the different control techniques, non-singular terminal sliding mode control represents an important control design for friction micromachines [23]. However, this technique is based on a well-identified friction model [23]. Techniques based on data training require many stages of experimentation for friction compensation design [6]. Other techniques require a pre-digital filter design, based on fitting system parameter data through experimentation [24]. Our control technique approach is not based on the mathematical model of friction or prolonged experimentation, and the simplicity of the obtained controller goes beyond that given by the sliding mode control theory. Therefore, our main contribution is a simple discontinuous control algorithm, based on an optimizing method, and applied to micromachine devices, capable of mitigating friction effects.

The organization of the rest of the document is as follows. Section two provides the description of the dynamic optimizer set in [8]. Section three presents our main contribution by adapting this optimizer to a regulation control scheme, together with numerical experiments. Mathematical proof of the closed-loop system is also provided by using Lyapunov’s theory. The application of our control scheme to a frictional micromachine device, along with numerical experiments, is given in Section four. Finally, sections five and six present the conclusions and future work, respectively.

## 2. The Benchmark Dynamic Optimizer

The goal of this section is to show the dynamic optimizer described in [8]. It is described in the following theorem:

**Theorem** **1**(Dynamical optimizer). *Given a smooth concave function f(x) with a single maximum point at x=a, and k being a given positive constant value, the dynamic optimizer given in Figure 1 produces a finite-time convergence of x(t) to a. That is, there exists value $T_{s}<\infty$, such that:*
(1)$$\lim_{t\to T_{s}}x\left(t\right)=a.$$

**Proof.** See [8]. □

Obviously, the following assumptions are inherent:There is only one maximum point of the function.The function is concave.

Moreover, the parameter *k* controls the speed of convergence of the optimizer output response. Finally, a numerical example is shown in Figure 2 with $f\left(x\right)=-{\left(x-1\right)}^{2}$, where the extremum of the function is located at x=1.

## 3. The Recent Control Scheme

The main objective of this section is to introduce our new approach to the regulation control statement using our dynamic optimizer, as seen in Figure 3. Finite-time convergence was also a goal. We considered the system to be stabilized was linear with an input and an output, and of a second order, given by: (2)$$\begin{array}{ccc} \dot{x}\left(t\right) & = & Ax(t)+bu(t), \end{array}$$(3)$$\begin{array}{ccc} y(t) & = & Cx(t), \end{array}$$
where $x\in R^{2}$ is the state vector of the system, and u∈R and y∈R are the input and output of the system, respectively. Additionally, the proposed model (Equation 2) and (3) is expressed in its canonical format:(4)$$A=\left[\begin{array}{cc} 0 & 1\\ \alpha_{1} & \alpha_{2} \end{array}\right],b=\left[\begin{array}{c} 0\\ \beta \end{array}\right],C=\left[\begin{array}{cc} 1 & 0 \end{array}\right],$$
and
(5)$$x\left(t\right)=\left[\begin{array}{c} x_{1}\left(t\right)\\ x_{2}\left(t\right) \end{array}\right].$$

Therefore, $\alpha_{1}$, $\alpha_{2}$, and β are the system parameters, and all are assumed to be constant values. On the other hand, we considered that the performance index f(x) was, instead, a convex function, and given by $f\left(x_{1}\left(t\right)\right)=\frac{1}{2}{\left(x_{1}\left(t\right)-a\right)}^{2}$. The parameter *a* was a given constant value. Hence, $\frac{\partial f(x_{1}\left(t\right)}{\partial x_{1}}=x_{1}\left(t\right)-a=y\left(t\right)-a$, as seen in Figure 3.

Next, is our main result.

**Theorem** **2**(Control scheme). *The closed-loop system, shown in Figure 3, was globally-stable having finite-time convergence of y(t) to a bounded set. That is, there existed a constant Ts<∞, such that:*
(6)$$\lim_{t\to Ts}|y\left(t\right)-a|=\Omega ,$$
*where Ω={y(t)∈R/∣y(t)−a∣≤δ}, and a∈R is a constant value representing the reference command signal, if the following conditions are satisfied:*
*There is a real constant value $k_{u}$ such that $\alpha_{1}-\beta k_{u}=-\epsilon_{1}<0$, with $\epsilon_{1}\in R^{+}$,**There is a real constant value $k_{d}$ such that $\alpha_{2}-\beta k_{d}=-\epsilon_{2}<0$, with $\epsilon_{2}\in R^{+}$,**There is a real constant value k such that $\beta k=-\epsilon_{0}<0$, with $\epsilon_{0}\in R^{+}$.*
*The next closed-loop system parameters setting was as follows:*

*The parameter $\epsilon_{2}$ was sufficiently large with respect to $\epsilon_{1}$ ($\epsilon_{2}>>\epsilon_{1}$),*

*The relation $\frac{\epsilon_{0}}{\epsilon_{1}}$ was sufficiently small ($\frac{\epsilon_{0}}{\epsilon_{1}}<<1$).*



**Proof.** From the block diagram shown in Figure 3, we could obtain:
(7)$$e\left(t\right)=y\left(t\right)-a=x_{1}\left(t\right)-a.$$Its time derivative, using the system model (Equation 2) and (3), yielded:
(8)$$\dot{e}\left(t\right)=\dot{y}\left(t\right)={\dot{x}}_{1}\left(t\right)=x_{2}\left(t\right).$$Once again, the time derivative of the above equation, using the system data (Equation 2) and (3), resulted in:
(9)$$\ddot{e}\left(t\right)={\dot{x}}_{2}\left(t\right)=\alpha_{1}x_{1}\left(t\right)+\alpha_{2}x_{2}\left(t\right)+\beta u\left(t\right).$$From Figure 3, we obtained:
(10)$$\begin{array}{cc} u(t) & =u_{a}\left(t\right)-k_{u}e\left(t\right)-k_{d}\dot{y}\left(t\right)\\ \; & =k\int sgn\left(e\left(t\right)\right)dt-k_{u}e\left(t\right)-k_{d}\dot{y}\left(t\right)\\ \; & =k\int sgn\left(e\left(t\right)\right)dt-k_{u}e\left(t\right)-k_{d}{\dot{x}}_{1}\left(t\right). \end{array}$$Then, Equation (Equation 10) into (Equation 9) produced:
(11)$$\ddot{e}\left(t\right)=\alpha_{1}x_{1}\left(t\right)+\alpha_{2}x_{2}\left(t\right)+\beta k\int sgn\left(e\left(t\right)\right)dt-\beta k_{u}e\left(t\right)-\beta k_{d}{\dot{x}}_{1}\left(t\right).$$Next, by taking into account that $x_{2}\left(t\right)=\dot{e}\left(t\right)$, $x_{1}\left(t\right)=e\left(t\right)+a$, and ${\dot{x}}_{1}\left(t\right)=\dot{e}\left(t\right)$, the equations are reduced to:
(12)$$\ddot{e}\left(t\right)=\left(\alpha_{1}-\beta k_{u}\right)e\left(t\right)+\left(\alpha_{2}-\beta k_{d}\right)\dot{e}\left(t\right)+\beta k\int sgn\left(e\left(t\right)\right)dt+\alpha_{1}a.$$Thereupon, if:
$\alpha_{1}-\beta k_{u}<0\to \alpha_{1}-\beta k_{u}=-\epsilon_{1}$; $\epsilon_{1}\in R^{+}$,$\alpha_{2}-\beta k_{d}<0\to \alpha_{2}-\beta k_{d}=-\epsilon_{2}$; $\epsilon_{2}\in R^{+}$,
then Equation (Equation 12) gives:
(13)$$\ddot{e}\left(t\right)=-\epsilon_{1}e\left(t\right)-\epsilon_{2}\dot{e}\left(t\right)+\beta k\int sgn\left(e\left(t\right)\right)dt+\alpha_{1}a.$$In the next place, the time-derivative of (Equation 13) shows:
(14)$$\dddot{e}\left(t\right)=-\epsilon_{2}\ddot{e}\left(t\right)-\epsilon_{1}\dot{e}\left(t\right)+\beta ksgn\left(e\left(t\right)\right).$$Later on, taking into account that $\beta k<0\to \beta k=-\epsilon_{0};\epsilon_{o}\in R^{+}$, we have:
(15)$$\dddot{e}\left(t\right)+\epsilon_{2}\ddot{e}\left(t\right)+\epsilon_{1}\dot{e}\left(t\right)+\epsilon_{0}sgn\left(e\left(t\right)\right)=0.$$Let us define:
(16)$$\ddot{z}\left(t\right)=\dot{e}\left(t\right)+\frac{\epsilon_{0}}{\epsilon_{1}}sgn\left(e\left(t\right)\right).$$Then, the system (Equation 15) yields:
(17)$$\dddot{e}\left(t\right)+\epsilon_{2}\ddot{e}\left(t\right)+\epsilon_{1}\ddot{z}\left(t\right)=0.$$Double time integration of (Equation 17) results in:
(18)$$\dot{e}\left(t\right)+\epsilon_{2}e\left(t\right)+\epsilon_{1}z\left(t\right)=0\to \dot{e}\left(t\right)=-\epsilon_{2}\left[e\left(t\right)+\frac{\epsilon_{1}}{\epsilon_{2}}z\left(t\right)\right].$$Now, if $\epsilon_{2}$ is large enough, the system (Equation 18) has a fast dynamic converging to:
(19)$$e\left(t\right)=-\frac{\epsilon_{1}}{\epsilon_{2}}z\left(t\right),$$
where z(t), the slow dynamic, is the solution to (Equation 16) by using (Equation 18) and (Equation 19):
(20)$$\ddot{z}\left(t\right)=\dot{e}\left(t\right)+\frac{\epsilon_{0}}{\epsilon_{1}}sgn\left(e\left(t\right)\right)\to \ddot{z}\left(t\right)=-\frac{\epsilon_{0}}{\epsilon_{1}}sgn\left(z\left(t\right)\right).$$System (Equation 20) represents a stable oscillator (See chapter 2 in [26]) with its amplitude proportional to the initial conditions of the system, and its oscillating frequency related to $\frac{\epsilon_{0}}{\epsilon_{1}}$. A numerical example is displayed in Figure 4 with $\frac{\epsilon_{0}}{\epsilon_{1}}=10$. The stability of this system (Equation 20) can be verified by invoking the next Lyapunov function:
(21)$$V\left(t\right)=\frac{\epsilon_{0}}{\epsilon_{1}}|z\left(t\right)|+\frac{1}{2}{\dot{z}}^{2}\left(t\right),$$
where its time derivative along the system trajectory (Equation 20) yields:
(22)$$\dot{V}\left(t\right)=0,\;\;a.e.,$$
implying stability of the cited system. This also concludes that all signals in the closed-loop system are bounded. Finite-time stability is assured, due to the convergence of the closed-loop system to its internal oscillator dynamic, and related to (Equation 19). □

Regarding proof development, it was assumed that the signum function is given by:(23)$$sgn\left(\cdot \right)=\left\{\begin{array}{cc} 1 & \mathrm{if}\;\cdot \geq 0\\ -1 & \mathrm{if}\;\cdot <0. \end{array}\right.$$

To give numerical examples, let us use the following system related to (Equation 2) and (3):(24)$$A=\left[\begin{array}{cc} 0 & 1\\ 1 & 1 \end{array}\right],b=\left[\begin{array}{c} 0\\ 1 \end{array}\right],C=\left[\begin{array}{cc} 1 & 0 \end{array}\right].$$

It was clear that this system was unstable. It could be verified that the conditions demanded by Theorem 2 were satisfied with $k_{d}=20$, $k_{u}=100$, and k=−2. On the other hand, in the derivative estimator, we set the time-constant τ=0.01. Figure 5 and Figure 6 showed the obtained numerical results. In both numerical experiments, the initial conditions were $x_{1}\left(0\right)=1.2$ and $x_{2}\left(0\right)=0$. The simulations were carried out using the application Xcos from the Scilab open source platform. Additionally, in this application, we used the integration method called RK45-Rungekutta(4/5). Obviously, the parameter *a* was used here as the command signal. It was assumed constant for control design only, and we demonstrated the performance of the controller by allowing it to be a time-varying signal. Additionally, a zoom in version of the Figure 5 is shown in Figure 7.

## 4. Application to a Micromachine Device

A schematic model of a piezoelectric positioning mechanism with friction is shown in Figure 8, where m=1.33 Kg, $b=9.5\times 10^{3}$ Ns/m, and $k_{m}=1.57\times 10^{6}$ N/m. See details in [1]. With these data, the system had the following information related to (Equation 2) and (3):(25)$$A=\left[\begin{array}{cc} 0 & 1\\ -1.18\times 10^{6} & -7.14\times 10^{3} \end{array}\right],b=\left[\begin{array}{c} 0\\ 0.75 \end{array}\right],C=\left[\begin{array}{cc} 1 & 0 \end{array}\right].$$

Additionally, the LuGre friction model (see Figure 8), was experimentally verified in [1], given by
(26)$$h\left(\dot{x}\left(t\right)\right)=\frac{13.1+7.3e^{-{\left(\dot{x}\left(t\right)-0.001\right)}^{2}}}{8.96\times 10^{5}},$$
(27)$$\dot{z}\left(t\right)=\dot{x}\left(t\right)-\frac{|\dot{x}\left(t\right)|z\left(t\right)}{h(\dot{x}\left(t\right))},$$
and
(28)$$F_{H}\left(t\right)=\left(8.96\times 10^{5}\right)z\left(t\right)+481.65\dot{x}\left(t\right)-480\frac{|\dot{x}\left(t\right)|z\left(t\right)}{h(\dot{x}\left(t\right))}.$$

On the other hand, $F_{L}\left(t\right)$ represented load variations and uncertainties due to modeling and external perturbations [1]. Here, in realizing our friction model, we used the estimation of the velocity information of the mechanism. Therefore, there existed model uncertainty, such that $F_{L}\left(t\right)$ was indirectly induced in our numerical experimentation. $K_{e}$ was the gain converting the applied voltage to the piezo-positioning micromachine to Newton force. Figure 9 shows the implementation of our controller for this friction micromachine using Scilab, as was previously invoked. Zero initial conditions on the plant was realized. Figure 10 shows the numerical experimental results. In Figure 9, notice that we used a gain block of $k_{a}=0.00001$ to scale for the readout. This same gain block for data acquisition is also called in [3]. Finally, to measure the performance of the control action, we performed the following time-error calculation in t∈[0,200] s:(29)$$M=\frac{\int_{0}^{T}|e\left(t\right)|dt}{T}=\frac{\int_{0}^{200}|e\left(t\right)|dt}{200},$$
yielding M=2.15 μm. Obviously, this value was in the micros-scale of micromachine devices. The maximum absolute error in the steady-state response was 0.41 μm. Compared to [1], the maximum absolute error obtained was approximately 0.5 μm for the tracking control case. Additionally, the control parameter $k_{d}$ affected the damping closed-loop dynamic behaviour, $k_{u}$ and *k* regulated the amplitude and the settling-time of the oscillating steady-state error signal. See Table 2 for a particular discussion. Finally, Figure 11 shows the final part of a long-term simulation.

Some additional comments. Mathematical modeling of piezo-positioning micromachines is not unique. First, because there are many friction models [7]. Second, the parametric identification is not unique for a given system [2]. Finally, a tracking error is observed in our numerical experiments, as well as in [3].

## 5. Future Work

Our control approach was based on the self-generation of a high frequency signal of small amplitude but without chattering. For some time, designing a sliding mode controller without chattering has been, amd still is, an important issue [27,28,29]. Our control scheme can be classified as a switching controller, because of the signum function where chattering is absent. Therefore, one option to improve our control performance is to incorporate more complex switching signals, as evidenced, for instance, in [30]. On the other hand, our control method implicitly generates the dynamics of an oscillator but its amplitude depends on its initial conditions. However, another option is to use an oscillating system with an asymptotically stable limit cycle, for example, as the one shown in [31,32]. Finally, another way to improve control performance is to add a friction compensator to the control law [33,34,35].

As future work, we aime to develop our control scheme by means of low cost electronics. This is because our control scheme can be fully assembled using only operational amplifiers (op-amps). Even the derivative estimator block in Figure 3 can be realized using op-amps. Op-amps are low-cost electronic units. Therefore, we have in mind to do this in our laboratory.

## 6. Conclusions

Based on a dynamic optimizer scheme with finite-time stability, a recent control approach was developed. In this approach, a self-oscillating signal was generated to meet the control statement objective. The control structure obtained seemed simple, and was even achievable through the use of analog electronics, such as operational amplifiers. In addition, the stability of the closed-loop system followed a recent point of view. Finally, controller performance was numerically evaluated by using a realistic LuGre friction model for micromachines.

## Figures and Tables

**Figure 1 micromachines-14-00387-f001:**
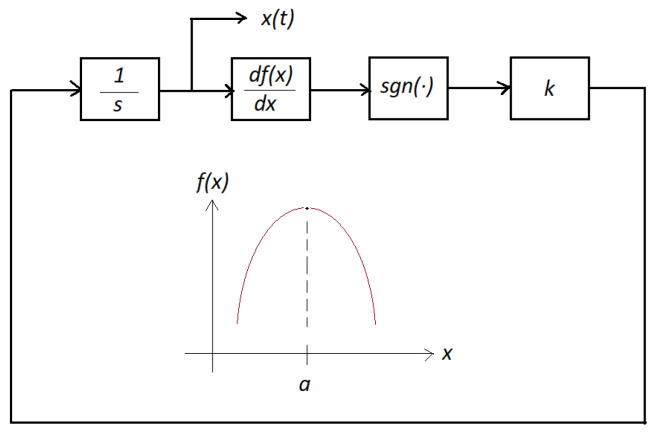
Dynamical optimizer.

**Figure 2 micromachines-14-00387-f002:**
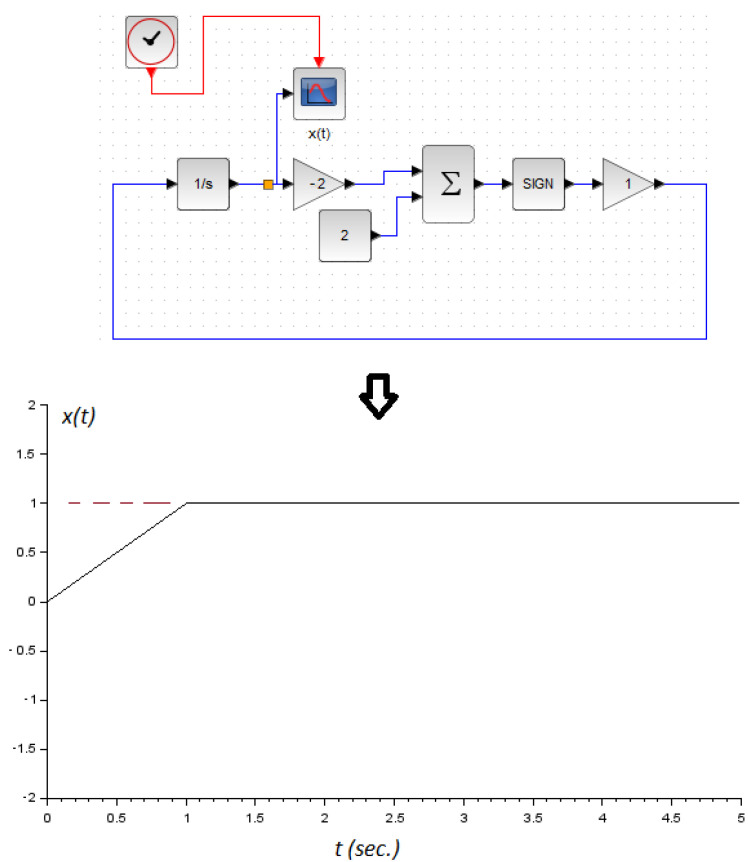
A numerical example using Xcos from Scilab application, where the numerical integration method used was RK54. The concave function is $f\left(x\right)=-{\left(x-1\right)}^{2}$. Note that the notation for the signum function sgn(·) in the Xcos environment is sign(·).

**Figure 3 micromachines-14-00387-f003:**
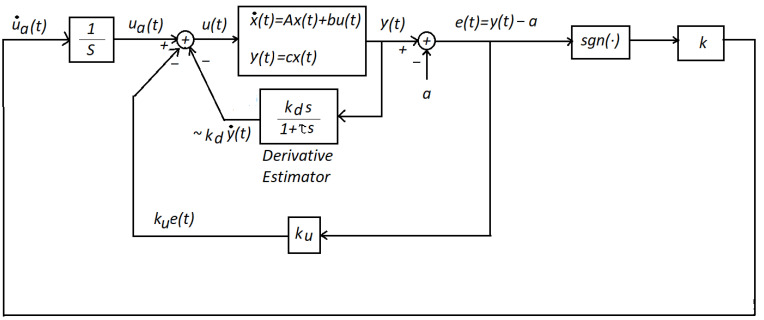
A recent control scheme using a dynamic optimizer. The parameter τ in the derivative estimator block was assumed to be small. This is a well-known derivative block (see, for instance, reference [25], p. 495).

**Figure 4 micromachines-14-00387-f004:**
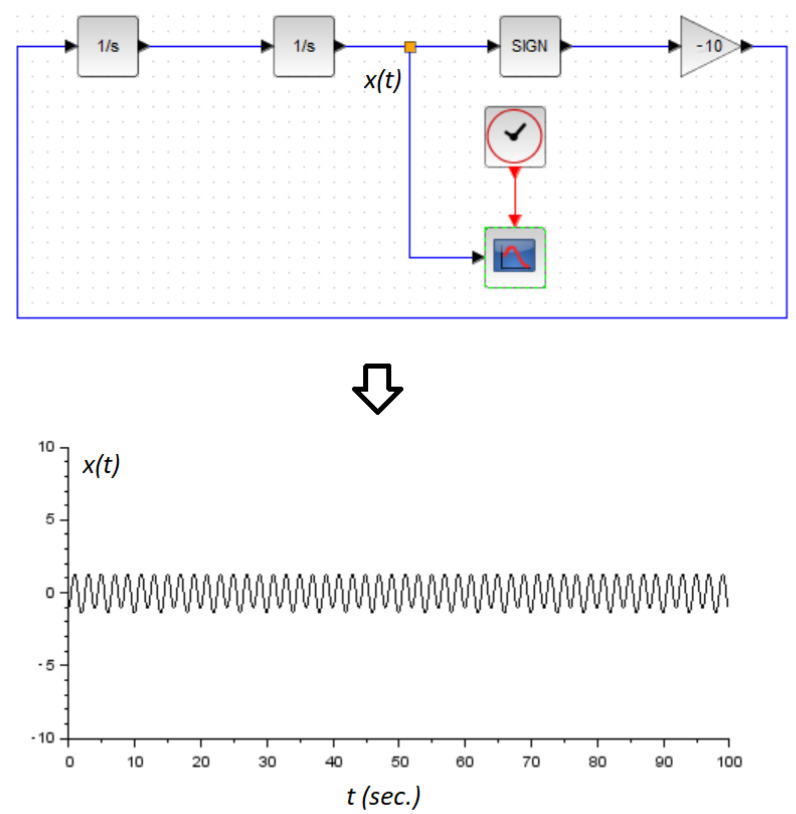
Numerical example of the oscillator dynamic (Equation 20). (**Top**): the *Xcos* block diagram of the system. (**Bottom**): the system response z(t) versus time. We use $\dot{z}\left(0\right)=1$ and z(0)=−1.

**Figure 5 micromachines-14-00387-f005:**
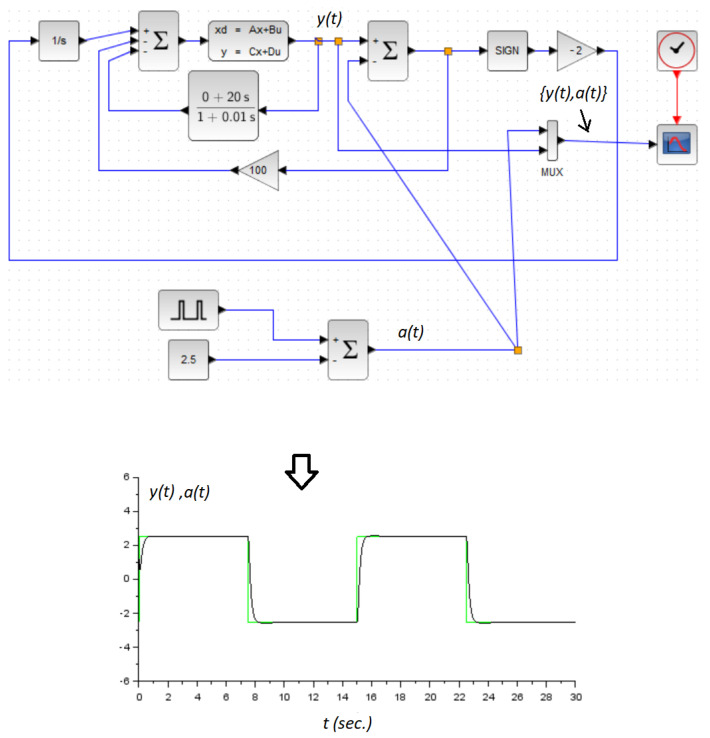
Numerical example: A pulse reference signal case. The green line is the reference signal a(t), and the black line is the system response y(t).

**Figure 6 micromachines-14-00387-f006:**
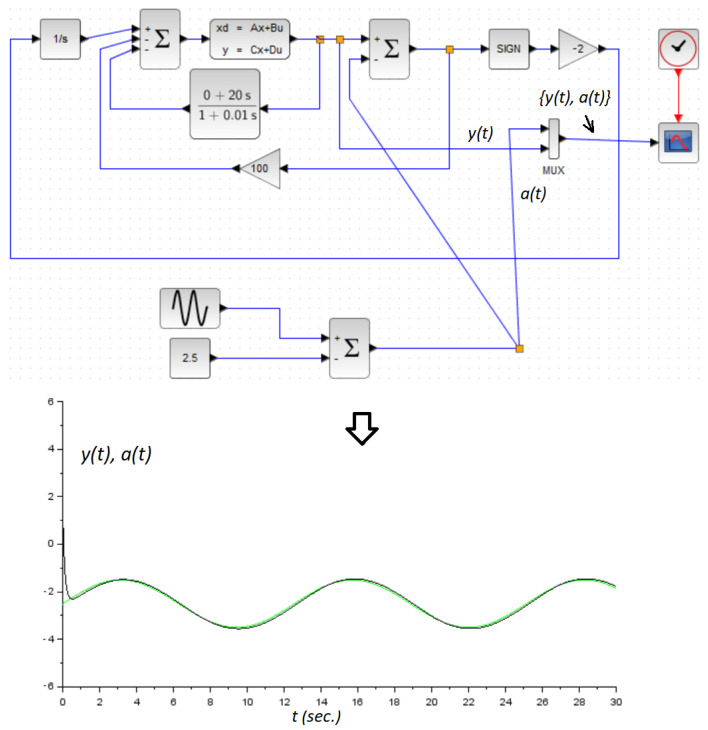
Numerical example: A sinusoidal reference signal case. The green line is the reference signal a(t), and the black line is the system response y(t).

**Figure 7 micromachines-14-00387-f007:**
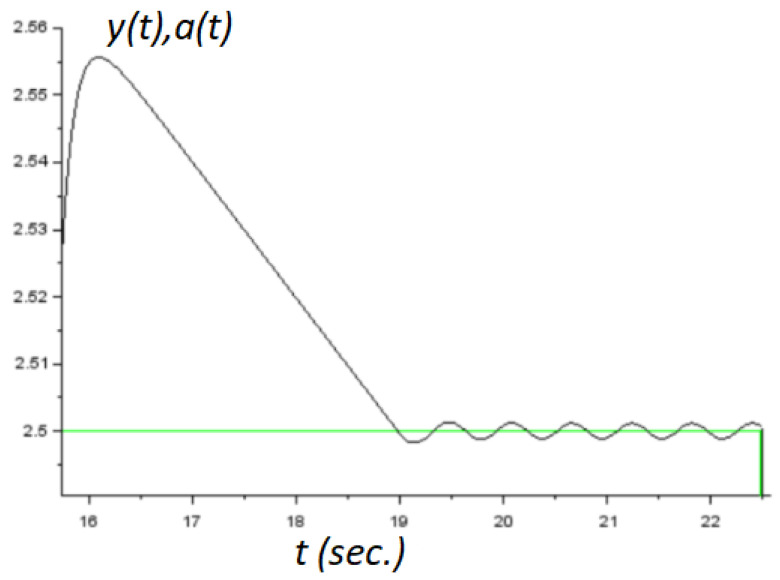
An enlarged view of the Figure 5.

**Figure 8 micromachines-14-00387-f008:**
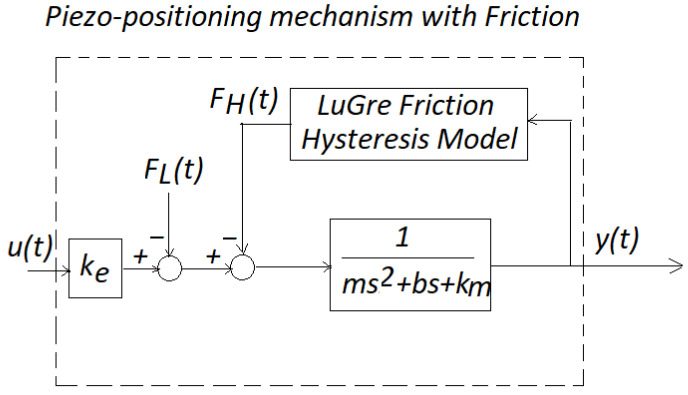
Model of a piezo-positioning mechanism with friction.

**Figure 9 micromachines-14-00387-f009:**
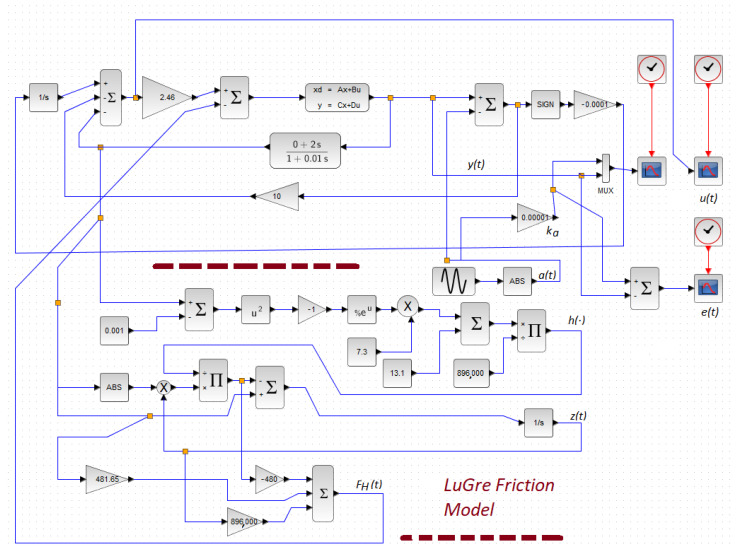
A control scheme for the positioning micromachine system using Xcos.

**Figure 10 micromachines-14-00387-f010:**
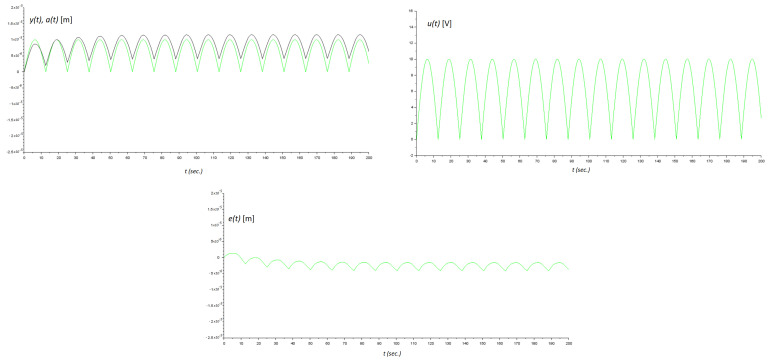
Results of numerical experiments. Top left: the green line is the reference signal a(t) and the black line is the position response of the micromachine y(t). Top right: control signal u(t). At the bottom: Error signal between system output and reference command e(t).

**Figure 11 micromachines-14-00387-f011:**
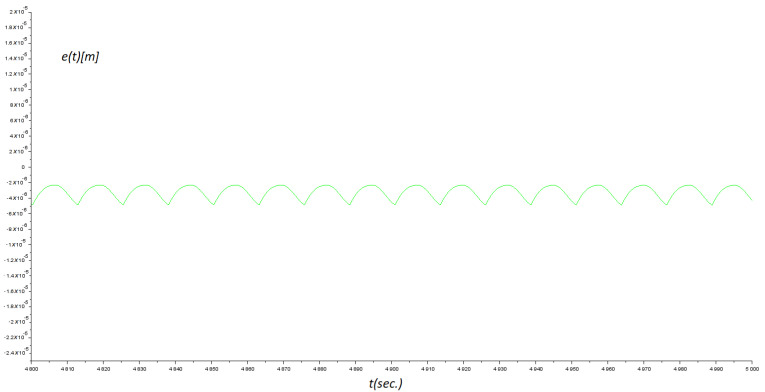
Results of numerical experiments. Error signal between system output and reference command e(t). The final part of a long-term simulation.

**Table 1 micromachines-14-00387-t001:** Main challenges for control design.

Challenge	Strategy	Evidence
Incorporate a dynamic optimizer in a closed-loop system	Search for a structure where the control signal is integrated to reduce the vibration that the optimizer and the plant could produce	Through numerical experiments
Closed-loop stability test	Invoke the theory of stability in the sense of Lyapunov	Verifying that the conditions of Lyapunov’s theory are met
To test control performance in a frictional micromachine using an experimentally validated system model	Use of numerical experiments	From numerical data, observe acceptable performance

**Table 2 micromachines-14-00387-t002:** Effects of the controller parameters on the dynamics of the closed-loop system (data obtained from numerical experimentation by independently increasing each control parameter).

Control Parameter	Steady-State Error	Transient Time Duration
*k*	–	↓
$k_{u}$	↑	↓
$k_{d}$	↑	–

## Data Availability

Data sharing not applicable.

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
