# Peer review of "A Control Method Based on a Simple Dynamic Optimizer: An Application to Micromachines with Friction"

_micromachines, 2023, doi:10.3390/mi14020387_

Round 1

Reviewer 1 Report (New Reviewer)

A regulation control method to mitigate the effects of friction is presented in the manuscript based on a simple dynamic optimizer. In the simulations, the LuGre friction model is adopted.

The topic falls in the journal scope. The manuscript is organized with tables and diagrams. The following concerns must be considered before it can be accepted for publication.

1.       Novelties of the paper should be better pointed out

2.       The model is applied to micro-machine devices but it is not clear where the small scale behaviour (since we deal with microstructures) enters.

3.       English should be checked.

Author Response

Thank you for your comments. Below is the reply to each of your remarks.

“A regulation control method to mitigate the effects of friction is presented in the manuscript based on a simple dynamic optimizer. In the simulations, the LuGre friction model is adopted.

The topic falls within the journal's scope. The manuscript is organized with tables and diagrams. The following concerns must be considered before it can be accepted for publication.”

Author’s reply: Thank you for your comments.

1.  Novelties of the paper should be better pointed out”

Author’s reply: The main contribution is the use of a dynamic optimizer in a closed-loop control system. See the orange text color on page 2 of the new paper version. In the author's opinion, this is a novelty in control design.

“2.  The model is applied to micro-machine devices but it is not clear where the small scale behavior (since we deal with microstructures) enters.”

Author’s reply: Thank you for the observation. The best option is to analyze the integral absolute error over a period of time. See the orange text color on page 10 of the new paper version.

“3.  English should be checked.”

Author’s reply: We have reviewed the text on paper and have modified it based on the support of a colleague. See the text in red in the new paper version.

Reviewer 2 Report (New Reviewer)

The paper presents a control method of a micromachine. Uses a LuGre friction model to represent friction effects. According to simulations, the model performance appears to be good. Notwithstanding, some metrics of the performance should be provided such as:

Maximum error, integral of the square error (over a period of the reference signal).

Also, some explanation about the values chosen for Kd, Ku, and K for this case should be provided. Is there a criterion? if this is the case, what is the criterion?

Author Response

Thank you for your comments. Below is the reply to each of your remarks.

“The paper presents a control method of a micromachine. Uses a LuGre friction model to represent friction effects. According to simulations, the model performance appears to be good. Notwithstanding, some metrics of the performance should be provided such as:

Maximum error, integral of the square error (over a period of the reference signal).

Also, some explanation about the values chosen for Kd, Ku, and K for this case should be provided. Is there a criterion? if this is the case, what is the criterion?”

Author’s reply: Thank you for your contribution. We have done an integral absolute error calculation over a period of time, and we have also said something about the controller parameters. Please refer to the orange text color in the new paper version.

Round 2

Reviewer 1 Report (New Reviewer)

Accept

Author Response

Thanks so much for your support!

Reviewer 2 Report (New Reviewer)

Although the manuscript is much clearer now, some issues remain:

The rewriting maintains some issues:

For example, in the Abstract: “…an important topic for your control…” Probably should be “…an important topic for a control…”.

In the Introduction (line 48): “This due to friction is highly…” Probably, it shouled be “This happens because friction is highly…”

An add other ones:

For example, again in the Abstract: “The the used optimizer…”

According to Figure 10, the setpoint oscillates between 0 and 10 µm (green line) and the position (black line) has an error increasing with time (black and green lines distance increases with time, apparently). Also, the average of the setpoint is circa 5 µm and the mean absolute error is 2.15 µm (more than 40%!). These results do not seem a good control system.

Please, explain or correct these results. And, by the way, I have asked for the mean square error not the mean absolute error.

Author Response

Thank you for your input. Below are the reply points.

1) “Although the manuscript is much clearer now, some issues remain:
The rewriting maintains some issues: For example, in the Abstract: “…an important topic for your control…” Probably should be
“…an important topic for a control…”.”

Author reply.- Thank you for this. It was corrected. See the new paper version in the text in red.

2) “In the Introduction (line 48): “This due to friction is highly…” Probably, it should be “This happens because friction is highly…””

Author reply.- Thanks. It was rewritten. See the new paper version in the text in red.

3) “An add other ones:
For example, again in the Abstract: “The the used optimizer…””

Author reply.- Thanks again. It was modified. See the new paper version in the text in red.

4) “According to Figure 10, the setpoint oscillates between 0 and 10 μm (green line) and the position (black line) has an error increasing with time (black and green lines distance increases with time, apparently). Also, the average of the setpoint is circa 5 μm and the mean absolute error is 2.15 μm (more than 40%!). These results do not seem a good control system.”

Author reply.- Thanks again. Obviously, in the simulation results, the transient response is also shown. However, in the steady-state behavior of the error signal, we can observe a maximum absolute error of 0.41 micrometers. And in comparison to the result shown in [1] for the tracking error case, it is observed that this error is approximately 0.5 micrometers.
These remarks are also given in the new paper version marked in red text color.

5) “Please, explain or correct these results. And, by the way, I have asked for the mean square error not the mean absolute error.”

Author reply.- Thank you. Very sorry. I didn´t understand your previous input because of the word ‘period’ in it. And to clarify, we slightly modified the equation (29) preceding the original formula. See the new paper version.

Round 3

Reviewer 2 Report (New Reviewer)

After two revisions, only one concern remains, but it is a big one:

The error increases with time, as it is clearly visible in Figure 10. The author has not corrected this issue and a control system whose error increases with time is useless. So, I must reject this manuscript for publication.

Author Response

I agree that the stability problem is an important part of any closed-loop system. Mathematically we have proved it. To clarify the simulation results, we repeat them for a long-term case. See the new Figure 11 in the new paper version, and the text in red. In this figure, we show the last part of this simulation, from 4800 to 5000 seconds.

Thank you.

This manuscript is a resubmission of an earlier submission. The following is a list of the peer review reports and author responses from that submission.

Round 1

Reviewer 1 Report

The dynamic optimizer designed in this paper is hardly innovative. The method has been proposed in [8].

From the abstract and the main content,  it is difficult to see that the paper is related to micromachines.

The figures in the paper is very poor, and it does not meet the standard of an academic paper at all

The simulation results cannot prove the effectiveness of the method.

Author Response

Thank you very much for your paper review. The following are our responses to each of your comments. 

Input: “The dynamic optimizer designed in this paper is hardly innovative. The method has been proposed in [8].”
Reply: Not precisely because using the dynamic optimizer within a control system requires math and control strategies, and validating the stability of the closed-loop system, as can be
seen in the main Theorem. This was not an easy task from a control design point of view. Therefore, our design presents a new control approach.

Input:”From the abstract and the main content, it is difficult to see that the paper is related to micromachines.
Reply: The abstract was modified. See the new paper version. For evidence, friction is an important topic in micromachines. See, for instance, reference [1].

Input:”The figures in the paper is very poor, and it does not meet the standard of an academic paper at all
Reply: Thanks a lot. Most of the figures were modified. Just to remind you, we are using Open Access software for simulations, and the quality of the software images is from a low-cost version. Sorry, and to facilitate data availability, we have decided to keep it as simple as possible for reference.

Input: “The simulation results cannot prove the effectiveness of the method.”
Reply: I do not agree. For example, reference [1] provides a complete validation of the friction model from Lugre model to Micromachine for simulation. And the control community,
there is a large set of simulation benchmarks to test new control strategies.

Best regards.

Reviewer 2 Report

The research is based on the numerical part, only in the simulation, comparative elements are needed, such as calculations or experimental ones, but, the following observations can be answered.

1.  The author should add a table in the first chapter and address the challenges and describe how this document addresses a challenge. "Introduction" section the discussion is not enough.

2.  It is recommended to present a comparative table with other articles or authors of the results.

3. Correctly apply the journal format

4. Add comparison of numerical results with theoretical calculations or with experimental data.

5. Improve image quality

6.Must update research references for at least the last 5 years.

Author Response

Thank you very much for your paper review. The following are our responses to each of your comments.

Input: ‘The author should add a table in the first chapter and address the challenges and describe how this document addresses a challenge. "Introduction" section the discussion is
not enough.
Reply: Thanks for your suggestion. See Table 1 in the new paper version.

Input:’It is recommended to present a comparative table with other articles or authors of the results.’
Reply: In the literature, it is difficult to find works on new control techniques by using a dynamic optimizer with a simple structure and applied to the control of a micromachine. Therefore, our contribution is a new approach.

Input:’Correctly apply the journal format
Reply: I think I have used the jour template in the latex environment. The one-column version was even invoked for large images.

Input:’Add comparison of numerical results with theoretical calculations or with experimental data.’
Reply: An experimental realization of the proposed control strategy may be future work. At this time, we have no experimental or numerical data to compare. Still, our main goals were met.

Input:’Improve image quality
Reply: Most of the figures were modified. Just to remind you, we are using Open Access software for simulations, and the quality of the software images is from a low-cost version. Sorry, and to facilitate data availability, we have decided to keep it as simple as possible for reference.

Input:’Must update research references for at least the last 5 years.’
Reply: At first I was trying to update some references but the structure of the article should be changed losing its presentation. Then, and as evidence, I found some recently published articles in the Micromachines Journal where some 'old' references were also used:
1) “AlScN Piezoelectric MEMS Mirrors with Large Field of View for LiDAR Application “
Yichen Liu , Lihao Wang, Yongquan Su, Yuyao Zhang, Yang Wang and Zhenyu Wu
Micromachines 2022, 13, 1550. https://doi.org/10.3390/mi13091550
Some references date from 2009,2004, and 2001.
2) “Design and Performance Analysis of LARMbot Torso V1”
Wenshuo Gao and Marco Ceccarelli
Micromachines 2022, 13(9), 1548; https://doi.org/10.3390/mi13091548and Marco Ceccarelli
Some references date from 2010,2008, and 1998.

Best regards. 

Reviewer 3 Report

1. The picture of the whole paper are very strange, and do not meet the requirements and quality of this journals. The author needs to make a comprehensive revision.

2. Similar as sliding mode control, the dynamic optimization method proposed in this paper also uses the symbolic function sgn(.). The simulation results show that the output has some flutter, so what is the advantage of this method compared with sliding mode control method. 

3. The micro positioning stage driven  by piezoelectric actuator usually uses flexure mechanism for transmission, and the flexure mechanism does not have friction. So, why does this paper uses the LuGre friction model to describe the piezoelectric driven micro position control, and what is the specific control object. "micromachines with friction" is not clearly defined.

4. Real control experiments need to be added to further verify the effectiveness of this method. The current content of the article is difficult to support this article to become a qualified journal paper.

Author Response

Thank you very much for your paper review. The following are our responses to each of your comments. 

Input: “The picture of the whole paper are very strange, and do not meet the requirements and quality of this journals. The author needs to make a comprehensive revision.”
Reply: Thank you. Most of the figures were modified. Just to remind you, we are using Open Access software for simulations, and the quality of the software images is from a low-cost version. Sorry, and to facilitate data availability, we have decided to keep it as simple as possible for reference.

Input: “Similar as sliding mode control, the dynamic optimization method proposed in this paper also uses the symbolic function sgn(.). The simulation results show that the output has some flutter, so what is the advantage of this method compared with sliding mode control method.
Reply: According to the given control structure, there is an integrator prior to the control signal to the plant. The integrator attenuates the vibration behavior of the signum function
block. Therefore, the induced vibration signal (the flutter) is produced for closed control stability. And this is also necessary for the correct operation of the optimizer block.

Input :”The micro positioning stage driven by piezoelectric actuator usually uses flexure mechanism for transmission, and the flexure mechanism does not have friction. So, why does this paper uses the LuGre friction model to describe the piezoelectric driven micro position control, and what is the specific control object. "micromachines with friction" is not clearly defined.”
Reply: According to reference [1], and depending on the structure of the micromachine, and its application, there is friction. Actually, we are using his LuGre friction model where the
friction parameters were obtained experimentally.

Input: “Real control experiments need to be added to further verify the effectiveness of this method. The current content of the article is difficult to support this article to become a
qualified journal paper.”
Reply: Yes, that's correct, partially. Since we do not have an experiment, we use a well-identified mathematical model of a friction micromachine which has been experimentally validated. Anyway, this can be future work. However, our control approach has its own position.

Best regards. 

Round 2

Reviewer 2 Report

Carry out future work to provide continuity to the research.

The implementation of the comments in the document is appreciated.

Author Response

Thanks for the suggestions. Some words are on paper. However, I have added the following text (see the new paper version, in red text):

“On the other hand, to experiment in future works, it is to develop our controller approach using a low-cost version. That is, if the micromachine is ready to use, including its position sensor and its forcing voltage amplifier (see Figure 8), the control scheme can be completely assembled using only operational amplifiers (op-amps). Just to recall, even the derivative estimator block in Figure 3 can be
realized using op-amps. Op-amps are low-cost electronic units. Therefore, we have in mind to do this in our laboratory. ”

Best regards.

Reviewer 3 Report

I don't think the picture quality of the revised version can reach the publishing level and quality of the journal. All coordinate pictures lack coordinate names and units.  The connections in Figure 9 are miscellaneous. I don't know why?

Author Response

Thanks for your comment. Now, most of the figures were modified again, including Figure 9 and others, adding coordinate and unit labels to the last one. See the new paper version. On the other hand, Scilab-Xcos is not as friendly as Matlab's Simulink is. So tracing the data buses required some skills in manipulating the computer mouse and clicking-menus for the data connection among various blocks.

Best regards. 

Round 3

Reviewer 2 Report

On my part, the work will provide a strong background for future research and it will be important to follow up on the research. I appreciate that you will take the recommendations into account.

Reviewer 3 Report

I have no other questions.